# Synthetic-to-Real Pose Estimation
# with Geometric Reconstruction

**Qiuxia Lin**[1]      **Kerui Gu**[1]      **Linlin Yang**[2,3]      **Angela Yao**[1]
[1]Department of Computer Science, National University of Singapore
[2]State Key Laboratory of Media Convergence and Communication, CUC
[3]School of Information and Communication Engineering, CUC
`{qiuxia, keruigu, ayao}@comp.nus.edu.sg`

## Abstract

Pose estimation is remarkably successful under supervised learning, but obtaining annotations, especially for new deployments, is costly and time-consuming. This work tackles adapting models trained on synthetic data to real-world target domains with only unlabelled data. A common approach is model fine-tuning with pseudo-labels from the target domain; yet many pseudo-labelling strategies cannot provide sufficient high-quality pose labels. This work proposes a reconstruction-based strategy as a complement to pseudo-labelling for synthetic-to-real domain adaptation. We generate the driving image by geometrically transforming a base image according to the predicted keypoints and enforce a reconstruction loss to refine the predictions. It provides a novel solution to effectively correct confident yet inaccurate keypoint locations through image reconstruction in domain adaptation. Our approach outperforms the previous state-of-the-arts by 8% for PCK on four large-scale hand and human real-world datasets. In particular, we excel on endpoints such as fingertips and head, with 7.2% and 29.9% improvements in PCK.

## 1   Introduction

Training deep models for human pose estimation, be it of the full body [30, 29, 2] or hands [37, 42], requires large-scale labelled datasets. Synthetic data [43, 31] has become more favoured due to the ease of generation and the challenges of accurately annotating real-world data. Despite advancements in realistic rendering [23, 5], models trained on synthetic data do not generalize well to real-world settings. The gap arises due to variations in pose, appearance, lighting, and other low-level differences between the two data domains. This paper aims to close this gap and targets learning from labelled synthetic and unlabelled real-world data for application to real-world settings.

Domain adaptation studies the transfer of models from synthetic to real data. Recent works on domain adaptive pose estimation [17, 36, 15] learn from target pseudo-labels. However, the gains from pseudo-labelling can be limited. There is an inherent contradiction – only labels which are sufficiently accurate benefit the learning, and it is important to ensure label quality through selection [15] and or correction schemes [36, 18]. Yet the ability to obtain high-quality pseudo-labels implies that the model is sufficiently capable of estimating the poses of said samples. The cross-domain setting makes it difficult to obtain high-quality labels. The challenge is compounded by additional gaps like different subjects' anatomical sizings, different annotation standards across datasets, etc.

Instead of estimating labels for the given target data, what if we try to reconstruct the data itself? This latter strategy is often used in self-supervised pose estimation, where learning is guided by reconstruction losses [33, 11, 34]. The challenge lies in linking the reconstruction scheme back to

the underlying pose[1]. While it is feasible to use a differentiable renderer, there are several drawbacks, such as the complexity of the renderer [23] and the difficulty of model fitting [16].

With this motivation, we design a simple yet effective reconstruction scheme that connects keypoint prediction via bone maps. Specifically, given a base and driving image pair with different poses, we estimate the keypoints and region-wise geometric transformations between the two poses, where each region corresponds to a bone linking two joints. The transformation is then applied with the predicted mask to obtain fine-grained warping (as shown in Fig. 1). The complete reconstruction is conducted in the generation module, which translates the base image into the driving image, specifically performing feature warping and inpainting in downsampling and upsampling structures. Intuitively, the more accurate the estimated keypoints and geometric transformation in the image pair, the better the reconstruction. Differences between the reconstruction and the original driving image can serve as a supervisory signal to improve keypoint localization.

A key benefit of the reconstruction scheme is that it can leverage almost *all* the samples in the unlabelled target dataset and give refinement according to the reconstruction loss. Pseudo-labeled samples are often selected based on some confidence measure like heatmap response, but given that most neural networks are poorly calibrated [22, 19], the confidence is often unsatisfactorily aligned with pose accuracy. Many samples can be highly confident yet inaccurate and the predictions remain inaccurate in this way. This is a key reason why pseudo-labelling methods easily saturate (see Table 3 (b)). In contrast, our reconstruction scheme requires only the two images containing the same object (hand/human) and a rigid background. As such, we are able to improve those confident but wrong keypoints based on 2D evidence in the target domain that arises from the reconstruction loss (see Table 3 (c)). Also, for keypoints that are both confident and accurate, the reconstruction will not degrade the result since it has perfectly provided useful information for reconstruction.

The supervision through reconstruction is indirect and is a weaker source of information than pose pseudo-labels. However, given that the two approaches are complementary, they can also be used in conjunction, *i.e.*, a limited but strong supervision from pseudo-labelling in combination with a larger and diverse source of weak supervision from the reconstruction. To that end, we additionally propose a new selection criteria for pseudo-labels that integrates 2D evidence and 3D kinematic constraints. In combination, we are able to outperform state-of-the-art domain adaptation methods by $8\%$, with significant improvements on difficult keypoints, like fingertips for the hand, and head for humans by $29.9\%$. Our contributions can be summarized as

- A novel reconstruction-based approach for domain adaptive pose estimation. We introduce a reconstruction method to make up for the limitations of the pseudo-labelling, and realize the connection between image reconstruction and keypoint prediction by building bone maps, which enables the correction of keypoints through optimizing reconstruction loss.

- An improved pseudo-labelling selection criteria that integrates both 2D evidence and 3D kinematic constraints. The new criteria improves the quality of the pseudo-labels by enlarging the set of usable pseudo-labels by 13%.

- Domain adaptation results from synthetic to real-world data for both hand and human body show that we outperform state-of-the-art by a large margin, with as much as 29.9% improvement on difficult keypoints like fingertips and head and an overall 8% improvement.

## 2 Related Work

### 2.1 Cross-domain Pose Estimation

Cross-domain challenges in pose estimation, especially in the context of synthetic-to-real, are best known for animal pose estimation [1, 20, 17], where it is challenging to collect and label sizable training data. Similar challenges, however, also exist for the human body [40] and hand [36, 18, 13], as deep learning algorithms become more data-hungry. The standard strategy is to apply consistency constraints [20, 36] or pseudo-labelling [17, 15] to learn effective features for the target domain. Consistency constraints can be applied in equivariant augmentations [20], and cross-modality [36].

---

[1]We make a distinction between data synthesis and reconstruction. The former generates image observations based on given poses; the latter recreates image observations without knowledge of the underlying pose.

To generate less-noisy pseudo-labels and guide the network fine-tuning, [15] utilized the normalized output from a teacher network and [1] proposed a progressive self-paced pseudo-label updating strategy to make the guidance more stable. Both works denoise pseudo-labels based on the heatmap activations while [36, 18] reduced noise with a pose correction procedure. Either strategy alone can lead to the neglect of additional correlation of depth or the accumulation of prediction errors. We therefore leverage both strategies simultaneously and embrace their union of pseudo-labels indicators when their consensus is high to enrich guidance on the target domain.

## 2.2 Conditional Image Reconstruction

Reconstructing images conditioned on paired images is an effective technique to concentrate on specific attributes (*e.g.*, keypoints) while ignoring irrelevant ones in pose estimation, where the focus can be summarized on conditions design [10, 11, 25]. Jakab *et al.* [10] proposed to disentangle keypoint heatmaps from appearance. The heatmaps, as conditions to generate images, can be optimized by the reconstruction loss. A follow-up work [11] utilized a skeleton image to facilitate the generation process. Furthermore, recent work [25] regressed affine parameters to deform a pre-defined shape template as conditions. However, they all extract appearance and keypoints from different images separately and overlook the geometric relationship across images, which can provide explicit guidance on image reconstruction.

Using geometric relationships across images to perform reconstruction has been seen in image animation [27, 26, 28]. They predict motion-specific keypoints [27, 26] or semantic regions [28], and obtain transformation parameters to deform one image to resemble another image. Those motion-specific keypoints or region-based methods are generalizable in predicting transformation parameters and modeling complex movements. But they target achieving the mean and deviation of each region, which is unrelated to our purpose of joint locations.

To emphasize the geometric relationship (*i.e.*, the transformations between the two images) to perform reconstruction for refining joint locations, we connect semantic regions and joint locations via bone maps, which enables the correction of poses through optimizing the reconstruction loss. Naturally, the reconstruction becomes a complement to pseudo-labelling, and both of them contribute to domain adaptive pose estimation.

# 3 Method

## 3.1 Problem Formulation

Consider labelled data $\mathcal{D}_s = \{(\mathbf{I}_{si}, \boldsymbol{y}_i)\}_{i=1}^{N_s}$ from a source domain $s$ and unlabelled data $\mathcal{D}_t = \{(\mathbf{I}_{ti})\}_{i=1}^{N_t}$ from target domain $t$, where $\mathbf{I}$ denotes an RGB image, $\boldsymbol{y}$ the pose label, and $N_s$ and $N_t$ are the number of samples in the source and target datasets. The objective is to learn a pose estimation system $f_{\text{kp}} = (f_{\text{enc}} \circ f_{\text{dec}})$, trained on $\{\mathcal{D}_s, \mathcal{D}_t\}$ and apply it to hold-out test data on the target domain $t$ for estimating 2D or 3D keypoints.

We follow other domain adaptive pose estimation methods, and pre-train on the labelled source domain (synthetic) and then fine-tune on the unlabelled target domain (real-world). In pre-training, (Sec. 3.2), we use the labelled synthetic data $\mathcal{D}_s$ to learn an initial pose estimation network. For fine-tuning on unlabelled real-world data $\mathcal{D}_t$ (Sec. 3.3), we introduce a pseudo-labelling strategy to correct 2D pseudo-labels via 3D constraints in Sec. 3.3.1. Additionally, we propose a geometric reconstruction to correct keypoint locations from image signals in Sec.3.3.2. The overall pipeline for fine-tuning is shown in Fig. 1.

## 3.2 Pre-training on the Source Domain

**Keypoint Predictor** The keypoint predictor $f_{\text{kp}} = (f_{\text{enc}} \circ f_{\text{dec}})$ is first pre-trained on the synthetic source images $\mathbf{I}_s$ to estimate a heatmap $\hat{h}_{\boldsymbol{p}}$ and a relative depth $\hat{d}$. As the synthetic images have ground truth labels, $f_{\text{kp}}$ can be learned in a supervised manner with ground-truths $(h_{\boldsymbol{p}}^{\text{gt}}, d^{\text{gt}})$:

$$\mathcal{L}_{\text{sup}} = \frac{1}{N_s} \sum_{\mathcal{D}_s} (\|\hat{h}_{\boldsymbol{p}} - h_{\boldsymbol{p}}^{\text{gt}}\|_2 + \lambda_d \|\hat{d} - d^{\text{gt}}\|_2), \tag{1}$$

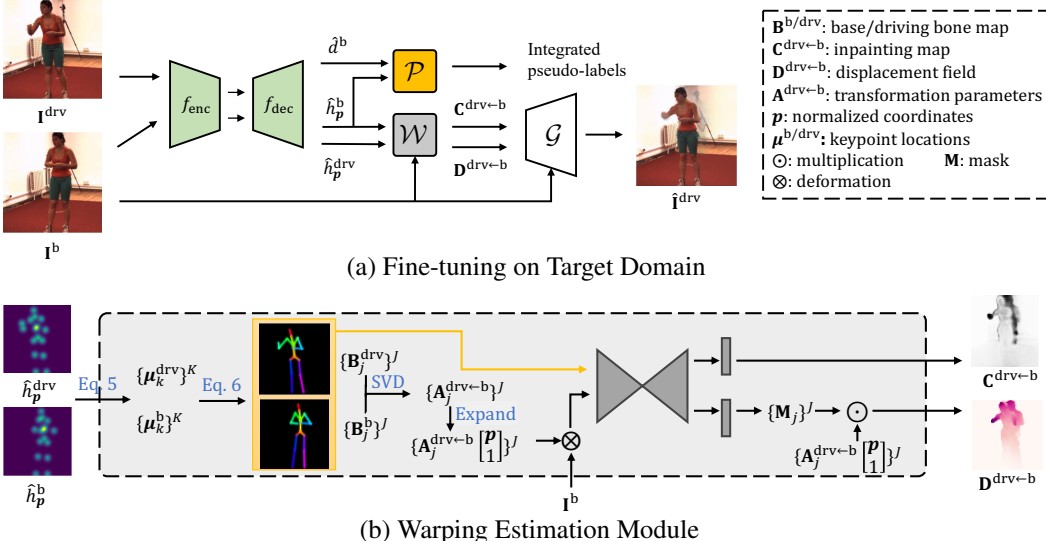

(a) Fine-tuning on Target Domain

(b) Warping Estimation Module

Figure 1: Illustration of pipeline and modules of fine-tuning. (a) Fine-tuning on target domain. Given the base and driving images sharing the same background but different poses, we forward them into keypoint predictor $f_{\mathrm{kp}} = (f_{\mathrm{enc}} \circ f_{\mathrm{dec}})$ to produce keypoint heatmaps $\hat{h}_{\boldsymbol{p}}$ and relative depth predictions $\hat{d}$. Afterward, our integrated pseudo-labelling $\mathcal{P}$ (See Sec. 3.3.1) is introduced to generate pseudo-labels, while the warping estimation module $\mathcal{W}$ and the generation module $\mathcal{G}$ (See Sec. 3.3.2) are used to reconstruct driving images from base images. (b) Warping estimation module. Bone maps derived from keypoints are used to calculate coarse transformation parameters via SVD and Expand. With the transformation and base images, a hourglass is used to generate inpainting map $\mathbf{C}^{\mathrm{drv} \leftarrow \mathrm{b}}$ and region-wise displacement field $\mathbf{D}^{\mathrm{drv} \leftarrow \mathrm{b}}$ that will work on upsampled feature maps in the generator $\mathcal{G}$.

where $\lambda_d$ is a hyper-parameter that trades off depth loss and heatmap loss. The ground-truth heatmap $h_{\boldsymbol{p}}^{\mathrm{gt}}$ is constructed as a 2D Gaussian centered at the ground-truth coordinate $(u^{\mathrm{gt}}, v^{\mathrm{gt}})$ and a standard deviation of 2. The ground-truth $d^{\mathrm{gt}}$ is the depth relative to the root keypoint. We follow the same practice in [9] to regress 2.5D and lift it to 3D pose. To elaborate, we employ soft-argmax operation on $\hat{h}_{\boldsymbol{p}}$ and then perform Hadamard product with depth value maps to derive the relative depth $\hat{d}$. The expected outcome of this operation yields the 2D coordinates, denoted as $(\hat{u}, \hat{v})$.

**Invariant Feature Learning** Models trained only on synthetic data $\mathcal{D}_s$ are unlikely to perform well on target data $\mathcal{D}_t$ (see Source only in Table 1). One reason is the domain gap between source and target data; typically, synthetic data used for pose estimation is not fully photorealistic, so the model does not generalize well to real-world data [4, 31]. To prevent $f_{\mathrm{kp}}$ from overfitting to source-specific representations, which is common [38, 21], we augment the source data to resemble target data and align the original and augmented source features using contrastive learning [3]. Specifically, we train a StyleNet from [24], which transfers the style of the target domain to the source, while preserving the source content, *i.e.*, the pose. The keypoint predictor $f_{\mathrm{kp}}$ is then trained with the supervised keypoint loss and an alignment on features with the following loss:

$$\mathcal{L}_{\mathrm{align}} = -\sum_{i=1}^{B} \log \frac{e^{(\langle \boldsymbol{z}_i, \boldsymbol{z}_i^+ \rangle / \tau)}}{\sum_{k=1}^{B} \mathbb{1}_{[k \neq i]} \big( e^{(\langle \boldsymbol{z}_i, \boldsymbol{z}_k \rangle / \tau)} + e^{(\langle \boldsymbol{z}_i, \boldsymbol{z}_k^+ \rangle / \tau)} \big)}, \tag{2}$$

with batch size $B$, temperature hyperparameter $\tau = 0.5$, and $\langle \cdot \rangle$ denotes the cosine similarity. In the above equation, the features $(\boldsymbol{z}, \boldsymbol{z}^+)$ are derived from the source and augmented source image.

[15] also applies a StyleNet to augment source samples but only supervises the network output. Our work extends the supervision to the latent features, as it can further benefit task-specific feature learning by building a consistent embedding space. It is feasible to also add target data into invariant feature learning with the style of source data, but we empirically find no obvious improvement. Therefore, we only perform alignment between source and augmented source features considering the limited memory.

### 3.3 Fine-tuning on the Target Domain

The fine-tuning on the target domain consists of integrated pseudo-labelling and geometric reconstruction. The integrated pseudo-labelling extracts confident pesudo-labels with guidance from both 2D heatmaps and 3D constraints. Since the label can still be noisy, we also have a geometric reconstruction strategy that generates a driving image based on the base image and pose transformation, which provides supervision simply from 2D image signals to improve those confident yet inaccurate keypoints.

#### 3.3.1 Integrated Pseudo-labelling

A commonly used strategy to adapt source models to a target domain is pseudo-labelling. In ensuring that pseudo-labels are valid, existing works have leveraged the heatmap activations in $\hat{h}_{\boldsymbol{p}}$ [15], or kinematic feasibility of the pose $\hat{\boldsymbol{y}}$ [36]. Both strategies have drawbacks; the former ignores correlations that may exist in $\hat{d}$, while the latter uses a greedy strategy that may accumulate errors down kinematic chains. With these considerations in mind, we propose an integrated pseudo-labelling strategy that combines the use of the 2D heatmap and the kinematic feasibility of 3D predictions. Specifically, given a predicted heatmap $\hat{h}_{\boldsymbol{p}}$, we generate a 2D confidence mask $\mathcal{M}_{2D} \in \mathbb{R}^{B \times K}$ as indicators to select keypoints with heatmap activations larger than the threshold $\gamma_{2D}$ in each batch during training. The 2D pseudo-label $\widetilde{h}_{\boldsymbol{p}}$ is the gaussian heatmap $\hat{h}_{\boldsymbol{p}}$ centered on the predicted 2D coordinate, as defined in [15]. For the 3D pseudo-labels, the prediction $\hat{\boldsymbol{y}} = (\hat{u}, \hat{v}, \hat{d})$ is lifted into 3D coordinates and rectified as pose $\widetilde{\boldsymbol{y}}$ with reasonable feasibility based on the pose correction [36]. The 3D confidence mask $\mathcal{M}_{3D} \in \mathbb{R}^{B \times K}$ indicates the qualified keypoints based on the comparison of L2 norm between the 2D projected coordinates before and after correction on $\hat{\boldsymbol{y}}$ to the threshold $\gamma_{3D}$.

For the first $T$ epochs of fine-tuning, the pseudo-labels $\widetilde{h}_{\boldsymbol{p}}$ and $\widetilde{\boldsymbol{y}}$ generated by 2D evidence and 3D correction, respectively, are both not confident, so we require them to train independently with their corresponding mask. Then we integrate the confidence mask $\mathcal{M}_{2D}$ and $\mathcal{M}_{3D}$ to complement each other and enrich the training set as $\mathcal{M} = \mathcal{M}_{2D} \cup \mathcal{M}_{3D}$, which considers more pseudo-labels for adaptation to both 2D and 3D. Overall, we optimize the following loss,

$$\mathcal{L}_{\text{pesudo}} = \begin{cases} \frac{1}{|\mathcal{M}_{2D}|} \sum_{\mathcal{M}_{2D}} \|\hat{h}_{\boldsymbol{p}} - \widetilde{h}_{\boldsymbol{p}}\|_2 + \frac{\lambda_y}{|\mathcal{M}_{3D}|} \sum_{\mathcal{M}_{3D}} \|\hat{\boldsymbol{y}} - \widetilde{\boldsymbol{y}}\|_2 & \text{epoch} < T \\ \frac{1}{|\mathcal{M}|} \sum_{\mathcal{M}} (\|\hat{h}_{\boldsymbol{p}} - \widetilde{h}_{\boldsymbol{p}}\|_2 + \lambda_y \|\hat{\boldsymbol{y}} - \widetilde{\boldsymbol{y}}\|_2) & \text{epoch} \geq T \end{cases}, \quad (3)$$

where $|\mathcal{M}|$ indicates the number of elements in $\mathcal{M}$ and $\lambda_y = 0.1$ is a coefficient that trades off pseudo-labelling between 2D heatmap loss and 3D loss. We find out that as fine-tuning proceeds, the overlap of confidence masks $\mathcal{M}_{2D}$ and $\mathcal{M}_{3D}$ will increase, at which point a single strategy can also provide valuable indicators for the other strategy.

#### 3.3.2 Geometric Reconstruction

Although the pseudo-labelling strategy is improved by utilizing both 2D and 3D information, the amount of pseudo-labels is limited and some confident labels are still incorrect. This is because these labels are generated from the model calibrated on the source domain, we therefore propose a geometric reconstruction that solely exploits specific features in the target domain. The performance of reconstructions is weaker compared to high-level pseudo-labeling. However, it can serve as complementary supervision by including all target data and correcting confident yet inaccurate keypoints by low-level evidence.

The geometric reconstruction consists of **Warping Estimation Module** $\mathcal{W}$ and **Generation Module** $\mathcal{G}$, as shown in Fig. 1(a). Given a pair of images from target domain $t$ containing human or hand $\{\mathbf{I}^b, \mathbf{I}^{drv}\}$ sampled in a video, The aim is to reconstruct driving image $\hat{\mathbf{I}}^{drv} \in \mathbb{R}^{3 \times H \times W}$ from base image $\mathbf{I}^b \in \mathbb{R}^{3 \times H \times W}$, where the image size is $H \times W$. To do so, the generation module $\mathcal{G}$ warps base image $\mathbf{I}^b$ based on specifications $\mathbf{D}^{drv \leftarrow b} \in \mathbb{R}^{\frac{H}{4} \times \frac{W}{4} \times 2}$ and $\mathbf{C}^{drv \leftarrow b} \in \mathbb{R}^{1 \times \frac{H}{4} \times \frac{W}{4}}$ estimated by the warping estimation module $\mathcal{W}$:

$$\hat{\mathbf{I}}^{drv} = \mathcal{G}(\mathbf{I}^b; \mathbf{D}^{drv \leftarrow b}, \mathbf{C}^{drv \leftarrow b}), \qquad \text{where } [\mathbf{D}^{drv \leftarrow b}, \mathbf{C}^{drv \leftarrow b}] = \mathcal{W}(\hat{h}_{\boldsymbol{p}}^b, \hat{h}_{\boldsymbol{p}}^{drv}). \qquad (4)$$

In Eq. 4, $\mathbf{D}^{\text{drv}\leftarrow\text{b}}$ and $\mathbf{C}^{\text{drv}\leftarrow\text{b}}$ denote a region-wise displacement field and an inpainting map respectively. The displacement field $\mathbf{D}^{\text{drv}\leftarrow\text{b}}$ is a dense vector field that specifies each pixel location in $\mathbf{I}^{\text{drv}}$ with its corresponding location in $\mathbf{I}^{\text{b}}$. And $\mathbf{C}^{\text{drv}\leftarrow\text{b}}$ is an attention map ranging from 0 to 1, where the $\mathcal{G}$ should do more inpainting on pixels with lower values. We apply a **Perceptual Loss** to penalize the semantic differences between the reconstructed image $\hat{\mathbf{I}}^{\text{drv}}$ and its original image $\mathbf{I}^{\text{drv}}$ to refine the keypoints. The details of each component (in bold) as follows.

**Warping Estimation Module** $\mathcal{W}$ Given the heatmap $h_{\boldsymbol{p}}$ estimated from $f_{\text{kp}}$, we decode it into coordinates by using a soft-argmax followed by an expectation; this combination is the standard approach followed in coordinate or integral regression methods [9, 29, 7, 6]. Specifically, we obtain the $k$-th 2D keypoint locations $\boldsymbol{\mu}_k$ in all $K$ keypoints while keeping the pretrained heatmaps:

$$\boldsymbol{\mu}_k = \sum_{\boldsymbol{p}\in\Omega} \boldsymbol{p}\cdot\tilde{h}_{\boldsymbol{p}}, \quad \text{where} \quad \tilde{h}_{\boldsymbol{p}} = \frac{e^{h_{\boldsymbol{p}}}}{\sum_{\boldsymbol{p}'\in\Omega} e^{h_{\boldsymbol{p}'}}}. \tag{5}$$

$\Omega$ is the area of the heatmaps, which are generated by $f_{\text{kp}}$.

Following the coarse-to-fine pipeline, to obtain a coarse motion of each body part, we need to obtain the pose transformation parameters. Previous works [25, 27] use the network to do direct regression but proved to have a poor generalization [28]. We thus follow [28] to use singular value decomposition (SVD) to compute transformation parameters of the bone map with respect to a reference heatmap, which is whitened and has zero mean and identity covariance. However, different from [28], where the bone maps are predicted by the network, we generate the bone map according to keypoint locations $\boldsymbol{\mu}_k$ and the kinematic tree. Formally, we assume there are $J$ chains in the kinematic tree and we denote the parent and child node of the kinematic chain $c_j$ as $\boldsymbol{\mu}_{jp}$ and $\boldsymbol{\mu}_{jc}$ respectively. The value of bone map $\mathbf{B}_j$ at pixel $\boldsymbol{p}$ of the $j$-th kinematic chain can be formulated as

$$b_j(\boldsymbol{p}) = e^{-d_j(\boldsymbol{p})/\sigma^2}, \tag{6}$$

where $\sigma$ is a hyperparameter that controls the thickness of the edge and $d_j$ denotes the distance between the pixel $\boldsymbol{p}$ and the edge determined by $\boldsymbol{\mu}_{jp}$ and $\boldsymbol{\mu}_{jc}$. Accordingly, the transformation parameters[2] $\mathbf{A}$ of $\mathbf{I}$ can be obtained by

$$\mathbf{A}_j = [\mathbf{U}_j\mathbf{S}_j^{\frac{1}{2}}, \boldsymbol{\mu}_j], \quad \text{where} \quad \underbrace{\mathbf{U}_j\mathbf{S}_j\mathbf{V}_j = \sum_{\boldsymbol{p}\in\Omega} b_j(\boldsymbol{p})(\boldsymbol{p}-\boldsymbol{\mu}_j)(\boldsymbol{p}-\boldsymbol{\mu}_j)^T}_{\text{SVD}}, \tag{7}$$

where $\boldsymbol{\mu}_j$ is the mean of $\boldsymbol{\mu}_{jp}$ and $\boldsymbol{\mu}_{jc}$. By Eq. 7, we can get the transformations $\mathbf{A}^{\text{b}\leftarrow\text{ref}}$ and $\mathbf{A}^{\text{drv}\leftarrow\text{ref}}$. Finally, the geometric transformation of bone $j$ from driving to base image is calculated by $\mathbf{A}^{\text{drv}\leftarrow\text{b}} = \mathbf{A}^{\text{drv}\leftarrow\text{ref}}(\mathbf{A}^{\text{b}\leftarrow\text{ref}})^{-1}$.

To generate a dense shift to deform $\mathbf{I}^{\text{b}}$ to approach $\mathbf{I}^{\text{drv}}$, we next convert $J$ transformation parameters to a fine-grained displacement field along with background transformation parameters. Specifically, with the input of $\mathbf{I}^{\text{b}}$, $\mathbf{B}^{\text{b}}$ and $\mathbf{B}^{\text{drv}}$, an hourglass network outputs $J+1$ masks $\mathbf{M}_j$ indicating one background and $J$ bones. $\mathbf{M}_0$ corresponds to the background. $\mathbf{M}_j(\boldsymbol{p})$ denotes the possibility of $\boldsymbol{p}$ belonging to the $j$-th region when $j > 0$. To ensure that the possibilities of every channel sum up to one, softmax is applied across the channels. Then, the displacement field $\mathbf{D}^{\text{drv}\leftarrow\text{b}}(\boldsymbol{p}) \in \mathbb{R}^2$, is generated by

$$\mathbf{D}^{\text{drv}\leftarrow\text{b}}(\boldsymbol{p}) = \sum_{j=0}^{J} \mathbf{M}_j(\boldsymbol{p})\, \mathbf{A}_j \underbrace{\begin{bmatrix} \boldsymbol{p} \\ 1 \end{bmatrix}}_{\text{Expand}}. \tag{8}$$

Through region-wise displacement field $\mathbf{D}^{\text{drv}\leftarrow\text{b}}$, the network gains insight into the specific locations on the base image that require deformation to match the driving image. Nonetheless, certain regions may be absent in the base image while present in the driving image. To identify these areas in need of inpainting by the network, we add an additional layer after the same hourglass to generate the inpainting map $\mathbf{C}^{\text{drv}\leftarrow\text{b}}$.

---

[2]We only need to rotate, translate, and scale to realize the transformation between the designed bone maps, which is different from the previous works based on affine transformation.

**Generation Module** $\mathcal{G}$   We build $\mathcal{G}$ with downsampling and upsampling, interconnected through skip connections. Similar to [28], the feature map from the skip connection is warped by displacement field $\mathbf{D}^{\mathrm{drv}\leftarrow\mathrm{b}}$ and then activated with inpainting map $\mathbf{C}^{\mathrm{drv}\leftarrow\mathrm{b}}$. We add the processed feature map to the $1 - \mathbf{C}^{\mathrm{drv}\leftarrow\mathrm{b}}$ activated feature map of the previous layer, and the sum is passed to the next layer. In this way, $\mathcal{G}$ can generate driving images through feature warping and limited inpainting from a similar image without the need for extensive rendering.

**Perceptual Loss** Following [14, 35], we apply a multi-scale perceptual $\ell_1$ loss with a pre-trained VGG-19 network. Mathematically, the reconstruction loss $\mathcal{L}_{\mathrm{rec}}$ is formulated as

$$\mathcal{L}_{\mathrm{rec}} = \sum_i |\phi_i(\hat{\mathbf{I}}^{\mathrm{drv}}) - \phi_i(\mathbf{I}^{\mathrm{drv}})|, \tag{9}$$

where $\phi_i(\mathbf{I})$ is the intermediate output of $i$-th layer of VGG-19.

**Training Procedure** The overall training of our framework is done in two stages. First is the pre-training on the source domain with supervised learning in Eq. 1 and invariant feature learning in Eq. 2 with weighting hyperparameter $\lambda_a$:

$$\mathcal{L}_{\mathrm{pre\text{-}train}} = \mathcal{L}_{\mathrm{sup}} + \lambda_a \mathcal{L}_{\mathrm{align}}. \tag{10}$$

The second stage finetunes in the target domain, with pseudo-labelling from Eq. 3 and geometric reconstruction from Eq. 9, together with supervision from the source domain to avoid over-fitting:

$$\mathcal{L}_{\mathrm{fine\text{-}tune}} = \mathcal{L}_{\mathrm{sup}} + \lambda_r \mathcal{L}_{\mathrm{rec}} + \lambda_p \mathcal{L}_{\mathrm{pseudo}}, \tag{11}$$

where $\lambda_r$ and $\lambda_p$ are balancing hyper-parameters. We have a warmstart in the fine-tuning where $\mathcal{L}_{\mathrm{sup}}$ and $\mathcal{L}_{\mathrm{pseudo}}$ are first used and then add $\mathcal{L}_{\mathrm{rec}}$ together to update the overall pipeline.

## 4   Experiments

### 4.1   Datasets & Evaluation Metric

**Hand Pose Datasets.**  For the source domain, we consider the synthetic dataset RHD [43] with 44k rendered images for training. For the unlabeled target domain, we consider real-world datasets H3D [41] and MVHand [39]; these have 11k/2k and 42k/42k training/testing splits respectively. For H3D, we consider only the subset with one-handed gestures. H3D and MVHand are both multi-view datasets; we treat each viewpoint as a "video" source.

**Human Pose Datasets.**  SURREAL [31] is a synthetic human dataset with 6 million annotations including keypoint locations. Human3.6M [8] annotates 3.6 million real-world indoor human poses. We follow protocol 1, using S1 and S5-S8 for training and S9 and S11 for testing. 3DPW [32] is a challenging outdoor dataset with 24 videos for training and 24 videos for testing. We obtain 2D keypoints by projecting the corresponding 3D ground truth.

**Evaluation Metric.**  The convention of domain adaptive pose estimation works [20, 15] is to evaluate 2D keypoint detection with Percentage of Correct Keypoint (PCK). We report PCK@0.05, which tabulates the percentage of correct predictions within the range of 5% of the input image size; a higher PCK indicates better performance. For 3D keypoint estimation, we evaluate with mean end-point-error (EPE); a lower EPE indicates better performance.

### 4.2   Implementation Details

Our framework is shown in Fig. 1. We use ResNet-101 as the feature extractor $f_{\mathrm{enc}}$ and deconvolutional layers as the decoder $f_{\mathrm{dec}}$ to produce a heatmap of size $2K \times 64 \times 64$ for 2D and depth. Each input image is cropped around the hand or the human and resized to $256 \times 256$. The initial learning rate is 1e-4, and we used Adam optimizer and degraded the learning rate with 0.1 at steps 60 and 75. Pre-training is done with a batch size of 64 for 40 epochs; fine-tuning is done with a batch size of 32 for 80 epochs. The hyperparameters $\lambda_a, \lambda_r, \lambda_p$ are set as 0.1. We compare with state-of-the-art domain adaptation methods: RegDA [12], CC-SSL [20], UniFrame [15], SemiHand [36], and DualNet [18]. Where possible, results are reported directly from the corresponding papers; otherwise, results are based on officially released code. In pre-training, RHD [43] and SURREAL [31] are

| Method | SURREAL→Human3.6M | | | | | | | | RHD→MVHand | | | | |
| --- | --- | --- | --- | --- | --- | --- | --- | --- | --- | --- | --- | --- | --- |

| Method | RHD→H3D | | | | | RHD→MVHand | | | | |
| --- | --- | --- | --- | --- | --- | --- | --- | --- | --- | --- |
| | MCP | PIP | DIP | Fin | All | MCP | PIP | DIP | Fin | All |
| Source only ($\mathcal{L}_{\text{sup}}$) | 67.4 | 64.2 | 63.3 | 54.8 | 61.8 | 32.1 | 70.5 | 64.6 | 39.8 | 52.7 |
| SemiHand [36] | - | - | - | - | 67.2 | - | - | - | - | 56.3 |
| DualNet [18] | - | - | - | - | 74.9 | - | - | - | - | 68.9 |
| CC-SSL [20] | 81.5 | 79.9 | 74.4 | 64.0 | 75.1 | - | - | - | - | 60.2 |
| RegDA [12] | 79.6 | 74.4 | 71.2 | 62.9 | 72.5 | - | - | - | - | 60.1 |
| UniFrame[15] | 86.7 | 84.6 | 78.9 | 68.1 | 79.6 | 45.0 | **77.7** | **71.0** | 57.6 | 62.9 |
| Ours | **89.4** | **88.3** | **81.9** | **73.9** | **84.1** | **76.4** | 76.9 | 68.7 | **61.0** | **70.8** |

Table 1: Hand 2D keypoint detection. Our method has the best average PCK@0.05 on both datasets and the most significant improvements on MCP and Fin.

| Method | SURREAL→Human3.6M | | | | | | | | SURREAL→3DPW | | | | | | | |
| --- | --- | --- | --- | --- | --- | --- | --- | --- | --- | --- | --- | --- | --- | --- | --- | --- |
| | Head | Sld | Elb | Wrist | Hip | Knee | Ankle | All | Head | Sld | Elb | Wrist | Hip | Knee | Ankle | All |
| Source Only | 51.3 | 69.4 | 75.4 | 66.4 | 37.9 | 77.3 | 77.7 | 67.3 | 40.9 | 59.2 | 65.6 | 64.2 | 64.4 | 56.4 | 39.8 | 57.9 |
| CC-SSL [20] | - | 44.3 | 68.5 | 55.2 | 22.2 | 62.3 | 57.8 | 51.7 | 41.0 | 58.5 | 76.3 | 67.4 | 77.6 | 61.7 | 47.8 | 62.5 |
| RegDA [12] | - | 73.3 | 86.4 | 72.8 | 54.8 | 82.0 | 84.4 | 75.6 | 41.8 | 65.8 | 73.3 | 68.1 | 72.8 | 65.0 | 49.3 | 64.8 |
| UniFrame[15] | 63.5 | 78.1 | 89.6 | **81.1** | 52.6 | 85.3 | **87.1** | 79.0 | 54.6 | 68.9 | 77.5 | 74.8 | 75.5 | 67.8 | 60.3 | 70.5 |
| Ours | **87.5** | **89.8** | **90.1** | 78.7 | **69.5** | **85.4** | 84.8 | **83.7** | **66.6** | **86.6** | **77.8** | **75.8** | **83.6** | **71.9** | **65.3** | **76.7** |

Table 2: The comparison on PCK@0.05 for 2D human keypoint detection. We achieve the best average performance on both datasets with the most significant improvements on the head.

used as labeled source dataset $\mathcal{D}_s$ in hand and human pose estimation, respectively. In fine-tuning, the training sets of H3D [41], MVHand [39], Human3.6M [8] and 3DPW [32] are individually used as unlabeled target dataset $\mathcal{D}_t$ for each pose adaptation experiment. And we evaluate on the corresponding test set.

### 4.3 Comparison with the State-of-the-Art

**Hand Pose.** Table 1 shows that for 2D keypoint detection on the hand, our method outperforms the previous best method by 5.7% and 12.6% on the H3D and MVHand datasets respectively. For more in-depth analysis, we also report the individual joint results averaged across the fingers *i.e.*, MCP, PIP, DIP, Fin[3]. Adapting *RHD → H3D* gives a consistent improvement, with increases of 3.1% to 8.5%. On *RHD → MVHand*, ours still achieves a significant improvement in PCK on MCP, increasing from 45.0 to 76.4.

Table 5 (b) shows that our method is state-of-the-art on 3D hand pose estimation for both *RHD → H3D* and *RHD → MVHand*. Especially notable is that we surpass DualNet [18], as we use only RGB images, while they incorporate additional modalities like background masks and depth images.

**Human Body Pose.** Table 2 shows, for *SURREAL → Human3.6M* and *SURREAL → 3DPW* that our method outperforms state-of-the-art UniFrame [15] by 5.9% for Human3.6M and 8.8% for 3DPW. The biggest improvements come in the head, shoulders, and hips. In particular, we observe the hip problem arising from the gap in annotation convention between SURREAL and Human3.6M, so all methods (including ours) have lower accuracy on the hips. However, we outperform others by 32.1% on Human3.6M, indicating our geometric reconstruction can help the model less affected by the annotation gap.

### 4.4 Analysis

**Effectiveness of Components.** We made two core contributions: geometric reconstruction and integrated pseudo-labelling. We here provide the gain of each component in Table 3 (a). $\mathcal{L}_{\text{UniFrame}}$ means we use the pseudo-labelling strategy in [15]. The comparison with $\mathcal{L}_{\text{UniFrame}}$ reveals the effectiveness of integrated pseudo-labelling with a 1.2 PCK improvement. Together with geometric reconstruction, we can further boost the increase to 2.7. $\mathcal{L}_{\text{heatmap}}$ means that we remove the warping estimation module and purely use heatmaps of both images to reconstruct $\hat{\mathbf{I}}^{\text{drv}}$. The results show that using explicit guidance in the generation module could help effectively link reconstruction back to the underlying poses.

---

[3]Hand joints moving away from the palm: MetaCarpoPhalangeal (MCP), PIP (Proximal InterPhalangeal), DIP (Distal InterPhalangeal) and Fin (Fingertips).

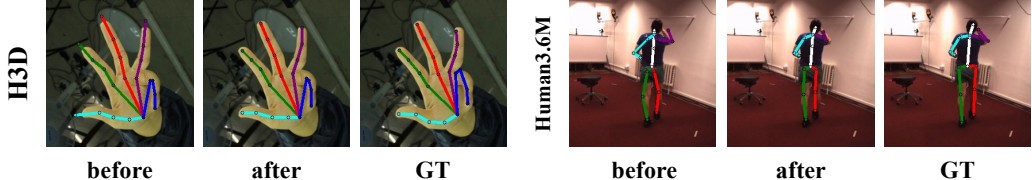

**before**    **after**    **GT**      **before**    **after**    **GT**

Figure 2: Comparisons before and after applying geometric reconstruction. Our method can correct the predictions that are on the background (left) and alleviate the annotation gap (right).

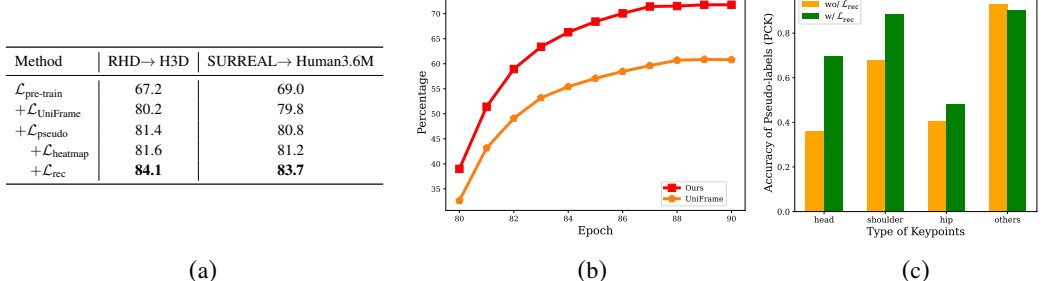

| Method | RHD→ H3D | SURREAL→ Human3.6M |
|---|---|---|
| $\mathcal{L}_{\text{pre-train}}$ | 67.2 | 69.0 |
| $+\mathcal{L}_{\text{UniFrame}}$ | 80.2 | 79.8 |
| $+\mathcal{L}_{\text{pseudo}}$ | 81.4 | 80.8 |
| $+\mathcal{L}_{\text{heatmap}}$ | 81.6 | 81.2 |
| $+\mathcal{L}_{\text{rec}}$ | **84.1** | **83.7** |

(a)        (b)        (c)

Table 3: (a) Ablation study on the proposed components and the reconstruction module contributes the most to the final performance. (b) Percentage of correct pseudo-labels in the training set of MVHand dataset. Both strategies saturate but the proposed integrated pseudo-labelling induces more correct labels. (c) Accuracy comparison before and after applying reconstruction loss $\mathcal{L}_{\text{rec}}$ for a specific set of confident keypoints on Human3.6M training set. After applying $\mathcal{L}_{\text{rec}}$, we are able to improve the pseudo-label of those keypoints previously proposed using only $\mathcal{L}_{\text{pseudo}}$ and $\mathcal{L}_{\text{sup}}$.

**Keypoint Correction by Geometric Reconstruction.** We analyze the impact of geometric reconstruction on confident yet inaccurate keypoints. We first train the model with $\mathcal{L}_{\text{sup}}$ and $\mathcal{L}_{\text{pseudo}}$ until convergence, and then save the indices of confident keypoints. Next, the reconstruction loss $\mathcal{L}_{\text{rec}}$ is added to fine-tune the model. We show the accuracy of the pseudo-labels for the keypoints associated with the saved indices, both before and after adding the reconstruction loss in Table 3 (c). The reconstruction loss obviously improves the confident yet inaccurate keypoints (*i.e.*, head, shoulder and hip), while having a negligible influence on other keypoints. This is because confident yet inaccurate keypoints will cause large reconstruction errors based on image evidence while others will not. Also, we observe that those confident yet inaccurate keypoints tend to be specific keypoints, and speculate it is due to the annotation gap.

**The Amount of Correct Pseudo-labels.** We select pseudo-labels with high confidence. However, we cannot assure that their accuracy is within PCK@0.05 for effective adaptation. We therefore tabulate the percentage of correct pseudo-labels, whose accuracy is within PCK@0.05, in unlabeled training data aggregated by the last ten epochs. As shown in Table 3 (b), the percentage saturates to 59% for the pseudo-labelling strategy and to 72% for the proposed integrated pseudo-labelling. Since the pseudo-labels are explored on training data with augmentations, their accuracy might not align consistently with the accuracy observed on the evaluation set. It is evident that both our approach and Uniframe eventually reach a saturation point for accurate pseudo-labels, but our method involves more usable pseudo-labels and leads to a better result.

**Fairness & Stride Strategy.** In our method, we need a pair of images with a similar background and the same object (human/hand). In contrast, existing state-of-the-art methods only require a single image during training on the target domain. For a fair comparison, we first consider the most common module to utilize adjacent paired input, *i.e.*, temporal consistency by applying $\ell_2$ loss on the predicted heatmaps $\|h_{\boldsymbol{p}}^i - h_{\boldsymbol{p}}^{i+1}\|_2$ of two consecutive frames $\mathbf{I}_i$, $\mathbf{I}_{i+1}$. Results in Table 5 (a) show it can only gain marginal improvement. Another consideration is whether the difference between the two poses affects the effectiveness of the reconstruction. We attempted to amplify the pose difference of two images from small to large by adjusting the stride from [0,5] to [30, 50]. As shown in Table 4, our method performs best on [5,15]. This makes sense that degenerately close

| Stride | | [0, 5] | [0, 15] | [5, 15] | [15, 30] | [30, 50] |
|---|---|---|---|---|---|---|
| SURREAL→ Human3.6M | | 81.8 | 83.5 | **83.7** | 83.1 | 82.5 |

Table 4: Ablation study on frame strides for the task of *SURREAL→Human3.6M* evaluated by PCK@0.05. We can see that all strides lead to improvement, and our method is robust to the selection of strides within a certain range.

| Method | RHD→MVHand | SURREAL→Human3.6M |
|---|---|---|
| RegDA[12] | 60.7(↑ 0.6) | 76.4(↑ 0.8) |
| UniFrame[15] | 63.4(↑ 0.5) | 79.3(↑ 0.3) |
| Ours | **70.8** | **83.7** |

(a)

| Method | RHD→ H3D | RHD→ MVHand |
|---|---|---|
| Source only | 27.77 | 21.21 |
| SemiHand[15] | 19.19 | 19.75 |
| DualNet[18] | 17.08 | 16.45 |
| Ours | **16.92** | **16.37** |

(b)

Table 5: (a) We add temporal consistency to ensure equivalence in the input information, which only brings minor improvement. (b) The comparison on EPE (mm) for 3D hand pose. By only improving 2D and applying a simple 3D pseudo-labelling strategy, we beat the state-of-the-art methods.

frames will make the network find the shortcut to self-reconstruction and far frames will make the generation too difficult to have a meaningful perceptual loss on keypoint locations.

### 4.5 Qualitative Results

**Hand Pose.** Fig. 2 (left) shows a common case in only using pseudo-labelling, where the fingertip is predicted on the background, although it is near the ground truth location. Applying our geometric reconstruction results in more accurate fingertip predictions. Besides, the wrong prediction of the pinky fingertip falling on the palm can also be rectified via the perceptual loss in our reconstruction module.

**Human Body Pose.** Fig. 2 (right) shows the comparison of our method on a Human3.6M sample before and after applying geometric reconstruction, where the shoulders are much lower in "before" compared to "GT". One possible cause is the difference in the annotation conventions across datasets, e.g. The annotation of the shoulders in SURREAL is closer to the chest, while Human3.6M localizes them at a reasonable shoulder position (See Supp.). Similar differences due to the annotation gap are reflected in the accuracy of head and hips, as shown in Table 2. Our method, however, can refine the pose prediction by leveraging the correlation between bone map and reconstruction, which is independent of the pose prior learned in the source domain. More analysis can be seen in Supp.

## 5 Conclusion

Pseudo-labelling is a common strategy for domain adaptive pose estimation which selects confident predictions to supervise the network. However, the number of accurate labels under this selection will saturate with the progress of training and some of the wrong predictions will remain unchanged. We hereby introduce a novel geometric reconstruction module to relieve this problem that aims to reconstruct the driving image according to keypoint locations and geometric transformations from the base image to the driving image. The incredible improvement especially comes from keypoints that generalize extremely poorly to target domain by only pseudo-labelling. We hope this will inspire future work to explore reconstruction-based methods to bridge the gap between source and target domains.

**Acknowledgments** This research / project is supported by the Ministry of Education, Singapore, under its MOE Academic Research Fund Tier 2 (STEM RIE2025 MOE-T2EP20220-0015).

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
