# Synthetic-to-Real Pose Estimation
# with Geometric Reconstruction

**Qiuxia Lin**[1]     **Kerui Gu**[1]     **Linlin Yang**[2,3]     **Angela Yao**[1]

[1]Department of Computer Science, National University of Singapore

[2]State Key Laboratory of Media Convergence and Communication, CUC

[3]School of Information and Communication Engineering, CUC

{qiuxia, keruigu, ayao}@comp.nus.edu.sg

## A    Model Architectures

The pose predictor $f_{kp}$ consists of a feature extractor $f_{enc}$ using ResNet-101 and a decoder $f_{dec}$ with three convtranspose2d - bn - relu blocks. For StyleNet $f_{sty}$, we follow UniFrame [4] and use Aladin [6]. The warping estimation module $\mathcal{W}$ is based on an hourglass with five conv3×3 - bn - relu - pool2×2 in the encoders and five upsample2×2 - conv3×3 - bn - relu blocks in the decoders. After the hourglass, there are two convs to respectively generate mask and inpainting map $\mathbf{C}^{drv \leftarrow b}$. The mask is used to generate region-wise displacement field $\mathbf{D}^{drv \leftarrow b}$. The generation module $\mathcal{G}$ is based on U-Net with five conv3×3 - bn - relu - avg pool2×2 blocks in the encoders and five upsample2×2 - conv3×3 - bn - relu blocks in the decoders. In $\mathcal{G}$, we use the Johnson architecture [3] with two down-sampling blocks, six residual-blocks and two up-sampling blocks. The design follows [7]. The inputs are the base image, displacement field, and inpainting map. It downsampled 4× and upsampled 4× to get the output, i.e. the reconstructed image. The generator is pre-trained with predicted keypoints before applying the geometric reconstruction module.

## B    Complementarity Analysis

In this section, we introduce how the two proposed components work separately and how they complement each other. We conducted the experiments on the task of *RHD* [12]→ *H3D* [11] for hand pose and *SURREAL* [8]→ *Human3.6M* [1] for human pose. Baseline refers to only applying the pseudo-labelling strategy in [4] along with the pre-training $\mathcal{L}_{pre\text{-}train}$ used in our method. As shown in Table A, integrated pseudo-labelling strategy and geometric reconstruction bring 1.2 and 1.4 improvements separately for the task of *RHD*→ *H3D*; Moreover, combining the two will boost the performance and gain an improvement of 3.9. The same phenomenon can also be observed in the results of the *SURREAL*→ *Human3.6M* task. Both results indicate that the two components are not contradictory to each other but even complement each other.

| Method | $RHD{\to}H3D$ | | | | | $SURREAL{\to}Human3.6M$ | | | | | | | |
|---|---|---|---|---|---|---|---|---|---|---|---|---|---|
| | MCP | PIP | DIP | Fin | All | Head | Sld | Elb | Wrist | Hip | Knee | Ankle | All |
| Baseline | 87.7 | 85.0 | 79.6 | 68.6 | 80.2 | 74.1 | 77.8 | 89.3 | 80.4 | 49.8 | 84.2 | 85.7 | 79.8 |
| +pseudo | 89.0 | 86.4 | 79.8 | 69.5 | 81.4 (+1.2) | 74.5 | 81.1 | 90.0 | **80.6** | 52.3 | 84.7 | 85.3 | 80.8 (+1.0) |
| +rec | 88.7 | 85.5 | 78.9 | 73.2 | 81.6 (+1.4) | 87.0 | 81.0 | **90.2** | 78.2 | 50.0 | 84.7 | **85.8** | 81.1 (+1.3) |
| +pseudo + rec | **89.4** | **88.3** | **81.9** | **73.9** | **84.1** (+3.9) | **87.5** | **89.8** | 90.1 | 78.7 | **69.5** | 85.4 | 84.8 | **83.7** (+3.9) |

Table A: Complementarity analysis of geometric reconstruction and integrated pseudo-labelling on hand pose (left) and human pose (right). Both experiments show that the improvement by simultaneously adding two components is much larger than the addition of the improvements of applying the components separately. It also indicates that the proposed two modules complement each other.

37th Conference on Neural Information Processing Systems (NeurIPS 2023).

| PCK(@0.05) | H3D | MVHand |
|---|---|---|
| $\mathcal{M}_{2D}$ | 80.2 | 63.7 |
| $\mathcal{M}_{3D}$ | 80.5 | 65.8 |
| $\mathcal{M}_{2D} \cup \mathcal{M}_{3D}$ | 80.1 | 64.7 |
| $(\mathcal{M}_{2D}, \mathcal{M}_{3D})$ | 80.8 | 67.3 |
| Ours | **81.4** | **68.2** |

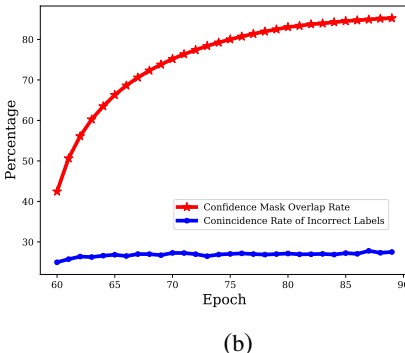

(a)                  (b)

Table B: (a) Ablation study of the combination strategies on 2D and 3D pseudo-labelling. The optimal strategy is to initially allow 2D and 3D pseudo-labelling to be independently trained using their respective mask. The masks can then be combined when there is a high overlap between them. (b) (Red line) The overlap of the confidence mask from 2D pseudo-labelling and 3D pseudo-labelling. The pseudo-labelling would become more confident and the overlap would be increased to 80% in the fine-tuning; (Blue line) The coincidence rate of incorrect labels for both the previous epoch $t-1$ and subsequent epoch $t$. Approximately 27% of the training data continuously selected as pseudo-labels is incorrect.

## C   Integrated Pseudo-labelling

### C.1   The Combination Strategies

As shown in Table B (a), we compared the performance of using only 2D pseudo-labelling with its mask $\mathcal{M}_{2D}$, only 3D pseudo-labelling with its mask $\mathcal{M}_{3D}$ and their combination. It can be observed that using 3D pseudo-labelling ($\mathcal{M}_{3D}$) arrives at a superior performance to 2D pseudo-labelling ($\mathcal{M}_{2D}$), which indicates that the kinematic constraints on 3D pose would greatly improve 2D results compared to the guidance from heatmap evidence. Then if directly using their combined mask ($\mathcal{M}_{2D} \cup \mathcal{M}_{3D}$) to guide the pseudo-labelling of 2D and 3D, it would degrade the performance compared to strategy $\mathcal{M}_{3D}$. This is because, at the beginning of fine-tuning, the domain shift is still large, and both 2D and 3D pseudo-labelling are not confident with only a 40% mask overlap rate, as shown by the red line in Table B (b). Independent learning ($(\mathcal{M}_{2D}, \mathcal{M}_{3D})$) could relieve that problem, but we found that the overlap of their mask would be over 80% in the later fine-tuning. And during the later fine-tuning, if we used the combination of their confidence mask, the PCK could be increased by at least 0.5 (80.8 to 81.4 and 67.3 to 68.2). This outcome highlights the capability of 2D and 3D pseudo-labelling techniques to mutually reinforce each other's confidence, leading to enriched pseudo-labels that would benefit the adaptation.

### C.2   Intrinsic Problems of Pseudo-labelling

We conducted analysis experiments on the MVHand [10] dataset to investigate pseudo-labeling, uncovering two underlying problems: saturation of correct labels and persistence of incorrect labels. Note that a pseudo-label is considered correct only if its accuracy is within the range of PCK@0.05.

In Sec. 4.4, we observed that the percentage of accurate pseudo-labels would saturate in the fine-tuning. Specifically, as the epoch increases, we accumulate the used pseudo-labels and calculate the percentage of pseudo-labels that have been correct at least once to the total training data. The experimental results show that even for our integrated pseudo-labelling, only 70% training data would be given as correct pseudo-labels throughout the fine-tuning, not to mention that this amount will be less per epoch. Besides, we calculated the coincidence rate of incorrect labels for both the previous epoch $t-1$ and subsequent epoch $t$. Assume that in $t$ epoch, the set of wrong pseudo-labels is $\mathcal{W}_t$, and $\mathcal{W}_{t-1}$ is the wrong pseudo-label set in epoch $t-1$, then we can have a coincidence rate of incorrect labels as $\frac{\mathcal{W}_{t-1} \cap \mathcal{W}_t}{\mathcal{W}_{t-1} \cup \mathcal{W}_t}$. As shown by the blue line in Table B (b), the coincidence rate remains constant at 27% as the epochs progress, indicating that more than a quarter of the data is continuously selected as pseudo-labels. However, these pseudo-labels are incorrect and consequently negatively impact the model adaptation.

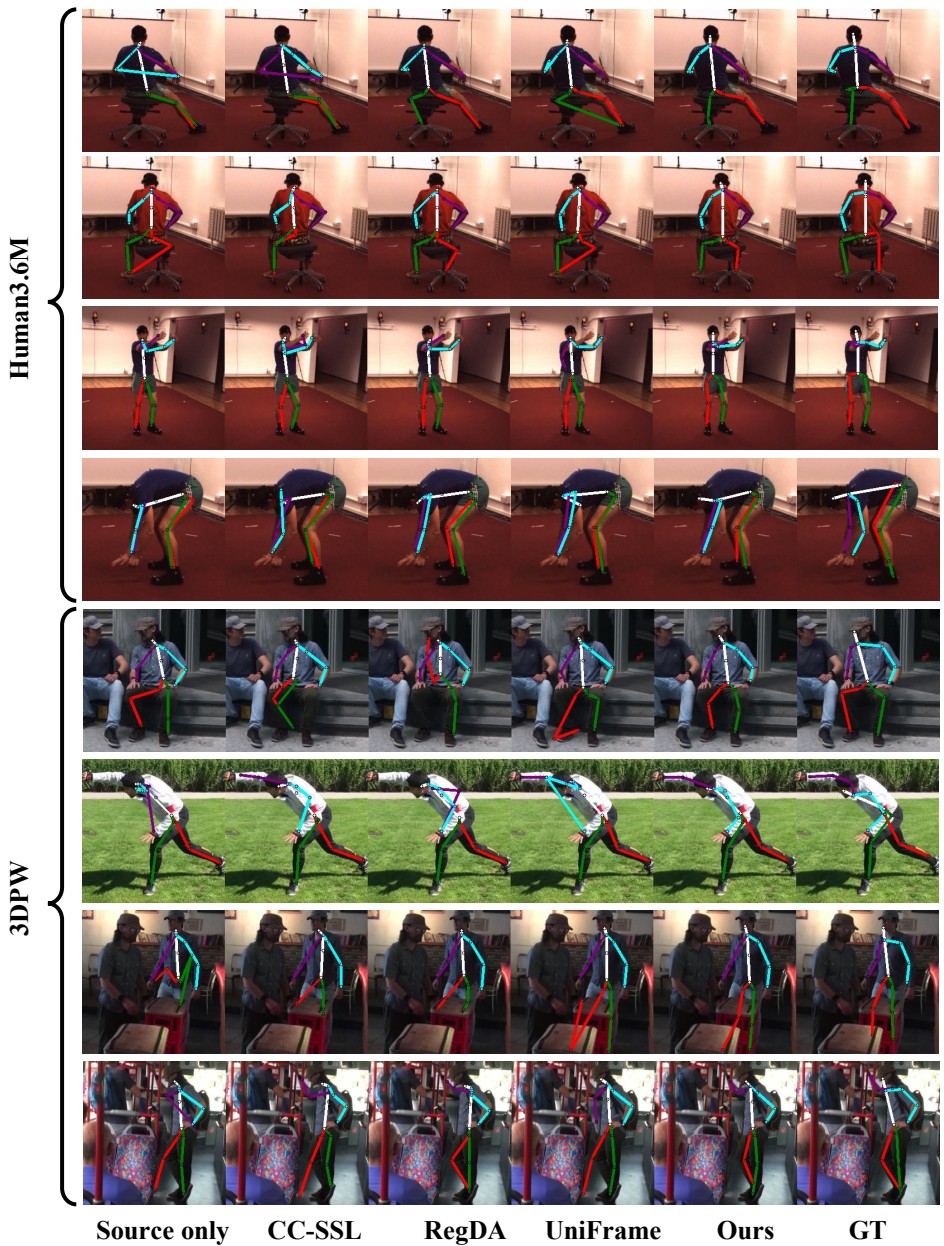

Figure A: Qualitative comparison of our method to state-of-the-art methods. The input images (from top to bottom) are from the Human3.6M [1] and 3DPW [9]. Comparing the other four methods, we can see that our method predicts better pose location in the presence of self-occlusion and object occlusions.

## D  Qualitative Results of Comparison

As shown in Fig. A, we provide the qualitative results of the adaptive pose estimation on two benchmarks. The comparisons are conducted with Souce-only, CC-SSL [5], RegDA [2] and UniFrame [4] methods. Our method outperforms others on pose accuracy. In particular, when self-occlusion occurs, other methods tend to predict the keypoints to other wrong keypoints or backgrounds. And these comparison methods also easily collapse when the person is occluded by objects in front of them (*i.e.*, a table). Compared to them, our method is less affected by the complex backgrounds and occlusions.

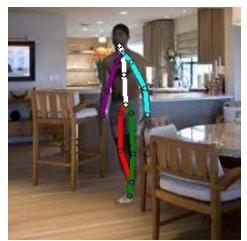 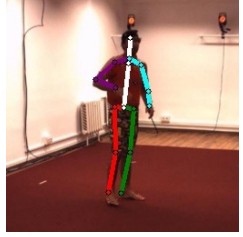 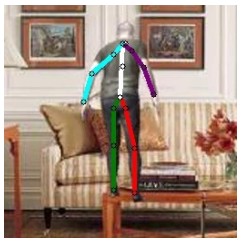 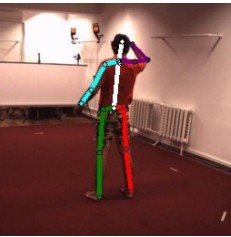

| **SURREAL** | **Human3.6M** | **SURREAL** | **Human3.6M** |

Figure B: Illustration of the annotation gap between a source domain (SURREAL) and a target domain (Human3.6M). The annotations of head, shoulders, and hips in the SURREAL dataset are very different from those in the Human3.6M. This annotation gap causes distribution-based or pseudo-labelling-based methods to perform poorly for these keypoints.

## E   Visualization of Annotation Gap

In the main paper, we claim a large annotation gap, especially in the human pose. Here, we provide some visual examples to illustrate in Fig. B. As shown, the annotation gaps obviously exist, especially for head, shoulders, and hips. Even some annotations in the source dataset are somewhat not accurate, for example, the ground-truth posterior arm (the link between shoulders and elbows) can include many background pixels. Previous domain adaptation pose estimation methods, including distribution-based RegDA, and pseudo-labelling-based UniFrame cannot address this problem effectively while our method alleviates it considerably.