# OpenReview forum: "Synthetic-to-Real Pose Estimation with Geometric Reconstruction"
_NeurIPS.cc/2023/Conference — NeurIPS 2023 poster_

### Official Review · Reviewer_mnmU · 2023-07-02

**Soundness:** 3 good
**Presentation:** 2 fair
**Contribution:** 2 fair
**Rating:** 5
**Confidence:** 4

**Summary:**

The authors proposed a geometric reconstruction-based domain adaptation method for synthetic to real transfer of hand/human pose estimation models. Based on the geometric transfer and image generation, the reconstruction loss can help the model generalize to the target domain. The proposed method can benefit from a wider range of unlabeled data compared to pseudo-label methods. The proposed method achieved improved performance on standard benchmarks.

**Strengths:**

1. Only part of the unlabeled data can be used for pseudo labeling while most images are suitable for the proposed reconstruction scheme.
2. The geometry-based reconstruction seems novel for hand/pose estimation, making the image generation easier and simplifying the training.
3. Quantitative results showed that the proposed method is quite effective for both hand and human pose estimation.

**Weaknesses:**

1. Reconstruction from pseudo-labels for domain adaptation is not entirely new. However, the model in this work used optical flow and occlusion maps for inpainting-based reconstruction, which seems to be quite effective without a complex reconstruction network. However, some implementation of the proposed model is not explained very clearly.
2. It’s unclear how much the perceptual reconstruction loss could help when predictions from the pretrained model are not very good. All results on reconstructions seem to be combined with pseudo labeling. For a sample with very inaccurate pose prediction, would the gradient from the perceptual reconstruction loss help? Or is a good pretrained model or a medium pretrained model + pseudo labeling necessary for the reconstruction module to work? Also see question 5.
3. Is the perceptual reconstruction only used in domain adaptation or also used to refine the predicted pose? For the results visualized in Figure 3, are they “before” or “after” the reconstruction fine-tuning of the model or something else?
4. Label cycle-consistency and refining predictions with the reconstruction were explored in related works and could also be of interest here.

**Questions:**

1. What is the architecture of the generator $\mathcal{G}$? Is the generator pretrained?
2. Is the legend text in Table 3 (b) reversed?
3. I thought in Table 3 (b) the performance should start from the pre-trained only performance in the training set (about 67.2 from Table 3 (a)) and improve. Why is the percentage of correct labels lower than 30% at the beginning of the adaptation?
4. What’s the limitation of the proposed reconstruction-based method? Does the adaptation collapse?
5. It seems that the authors used a different style transfer network than in UniFrame. Does the reconstruction-based adaptation work without the style transfer network or a worse initialization with a low percentage of the good pose estimations at the beginning?

**Limitations:**

1. See weakness 1: the idea of reconstruction is not entirely new.
2. See question 1,3,4: the clarity of the writing can be improved.
3. See weakness 2 and question 5: would be good to analyze the limitations of the model in other settings.

---

> ### Author Rebuttal · Authors · 2023-08-08
>
> **Q1: Reconstruction from pseudo-labels for domain adaptation is not entirely new. However, the model in this work used optical flow and occlusion maps for inpainting-based reconstruction, which seems to be quite effective without a complex reconstruction network. However, some implementation of the proposed model is not explained very clearly.**
>
> A1: Thanks for pointing this out. The pixel-wise flow predictor $\mathcal{P}$ is based on an hourglass with five conv3×3 - bn - relu - pool2×2 in the encoders and five upsample2×2 - conv3×3 - bn - relu blocks in the decoders. After the hourglass, there are two convs to respectively generate masks and occlusion map. The masks are used to generate dense optical flow $\rm{O}^{\text{drv}\leftarrow\text{b}}$ (See Eq.8). The design of generation module can be seen in A5. The pose predictor $f_{\text{kp}}$ consists of a feature extractor $ f_{\text{enc}}$ using ResNet-101 and a decoder $ f_{\text{dec}}$ with three convtranspose2d - bn - relu blocks.
>
> We will provide more implementation details in the Supplementary, and release source code.
>
> **Q2: It’s unclear how much the perceptual reconstruction loss could help when predictions from the pretrained model are not very good. All results on reconstructions seem to be combined with pseudo labeling. For a sample with very inaccurate pose prediction, would the gradient from the perceptual reconstruction loss help? Or is a good pretrained model or a medium pretrained model + pseudo labeling necessary for the reconstruction module to work? Also see Q11.**
>
> A2: Thanks for this interesting question. The reconstruction module can still bring medium-trained models to comparable results against the state-of-the-art methods without pseudo-labelling.
>
> Experimentally, we adapt SURREAL to Human 3.6M and apply only the perceptual reconstruction loss to a checkpoint model of 52.1, which has a really bad performance. After running 50 epochs, PCK reaches 74.8, which is a competitive result against state-of-the-art methods.
>
> **Q3: Is the perceptual reconstruction only used in domain adaptation or also used to refine the predicted pose? For the results visualized in Figure 3, are they “before” or “after” the reconstruction fine-tuning of the model or something else?**
>
> A3: The perceptual reconstruction is only used in domain adaptation. Yes, Figure 3 shows the prediction “before” or “after” the geometric reconstruction fine-tuning.
>
> **Q4: Label cycle-consistency and refining predictions with the reconstruction were explored in related works and could also be of interest here.**
>
> A4: Thank you for pointing this out. We will add more literature about label cycle-consistency [1,2] and refining predictions [3,4] in the related work.
>
> [1] DualPoseNet: Category-level 6D Object Pose and Size Estimation Using Dual Pose Network with Refined Learning of Pose Consistency, ICCV’21
>
> [2] Self-Supervised Geometric Correspondence for Category-Level 6D Object Pose Estimation in the Wild, ICLR’23
>
> [3] Learning to Refine Human Pose Estimation, CVPR’18
>
> [4] Pose-Oriented Transformer with Uncertainty-Guided Refniement for 2D-to-3D Human Pose Estimation, AAAI’23
>
>
> **Q5: What is the architecture of the generator $\mathcal{G}$? Is the generator pretrained?**
>
> A5: The generator is based on U-Net with five conv3×3 - bn - relu - avg − pool2×2 blocks in the encoders and five upsample2×2 - conv3×3 - bn - relu blocks in the decoders. In the generator network, we use the Johnson architecture [1] with two down-sampling blocks, six residual-blocks and two up-sampling blocks. The design follows [32]. The inputs are the base image, dense optical flow, and occlusion map. It downsampled 4x and upsampled 4x to get the output, i.e. the reconstructed image.
>
> Yes, the generator is pretrained with predicted keypoints before applying the geometric reconstruction module.
>
> Thank you for pointing out and we will include the above information in the final version.
>
>
> **Q6: Is the legend text in Table 3 (b) reversed?**
>
> A6: Thank you for pointing this out. Yes, they are reversed. We will correct them in the revision.
>
> **Q7: I thought in Table 3 (b) the performance should start from the pre-trained only performance in the training set (about 67.2 from Table 3 (a)) and improve. Why is the percentage of correct labels lower than 30% at the beginning of the adaptation?**
>
> A7: Sorry for the confusion. The plot given in Table 3(b) is conducted on MVHand while Table 3(a) shows results on H3D and Human3.6M. We will label this clearly in the revision.
>
> Note, for the plot in 3(b), the percentage of correct pseudo-labels at the beginning only represents the performance of partly training samples as the data loader cannot iterate the whole training samples in one epoch, which is keeping in line with the training scheme of Uniframe.
>
>
> **Q8: What’s the limitation of the proposed reconstruction-based method?**
>
> A8: One limitation is that our method cannot handle reconstruction ambiguity in intermediate keypoints, e.g. consider the case where the entire arm is straight, the location of the elbow is challenging for the reconstruction model.
>
> **Q9: Does the adaptation collapse?**
>
> A9: No, the adaptation will not collapse.
>
> **Q10: It seems that the authors used a different style transfer network than in UniFrame.**
>
> A10: No, we use Aladin [1] as the style transfer network, which is the same as UniFrame.
>
> [1] Aladin: all layer adaptive instance normalization for fine-grained style similarity, ICCV’21
>
> **Q11: Does the reconstruction-based adaptation work without the style transfer network or a worse initialization with a low percentage of the good pose estimations at the beginning?**
>
> A11: Please see A2.

---

> > ### Comment · Reviewer_mnmU · 2023-08-17
> > **Re: Rebuttal by Authors**
> >
> > Thanks the authors for the detailed response to my questions. All my concerns are addressed and I don't have any further questions.

---

### Official Review · Reviewer_RVKB · 2023-07-03

**Soundness:** 3 good
**Presentation:** 3 good
**Contribution:** 3 good
**Rating:** 5
**Confidence:** 4

**Summary:**

This paper proposed a domain adaptation method from synthetic to real for 3D hand/human pose estimation. Specifically, two self-supervised losses are designed, one is a geometric reconstruction loss which measures a multi-scale perceptual loss between an image in target domain and a generated one with pairwise geometric constraint, another is an integrated pseudo-labeling loss which measures integrated deviations on both 2D heatmaps and 3D poses. Experiment has been done on a set of benchmark dataset and better performance has been achieved compared with existing methods.

**Strengths:**

The proposed affine transformation module for optical flow generation and image reconstruction is interesting. The paper is easy to follow and the experiment result is promising. The ablation study clearly demonstrates the effectiveness of the paper's technical contributions.


**Weaknesses:**

1. One question is on the occlusion map described in Sec 3.3.2, it is not clear whether this occlusion map represent the occlusion between hand/human and background? or the self-occlusions between fingers/palm? does the occlusion map predictor uses the same hourglass network backbone as the 'Pixel-wise Flow Predictor'? whether the proposed geometric reconstruction module can handle self-occlusion?
2. Some details of comparison experiment maybe needed, i.e. whether the methods listed in table 1&2 use the same backbone network? is the computational cost and memory cost comparable, etc.

**Questions:**

Please ref to the weakness part

**Limitations:**

I donot see any potential negative societal impact

---

> ### Author Rebuttal · Authors · 2023-08-08
>
> **Q1: Does the occlusion map in Sec 3.3.2, represent the occlusion between hand/human and background? or the self-occlusions between fingers/palm?**
>
> A1: The occlusion does not represent either. Instead, it marks out regions that should be inpainted by the base image, and is applied as a mask to diminish the impact of the existing features in these regions. As as you can see the occlusion map in Fig. 2, the base image needs to inpaint the fingers to background pixels to match the driving image, so they are black. However, the palm does not need to inpaint, so they remain almost white.
>
> **Q2: Does the occlusion map predictor uses the same hourglass network backbone as the 'Pixel-wise Flow Predictor'?**
>
> A2: Yes, it uses the same backbone, though we add one more convolution head. Please refer to the architecture details of hourglass in R4 Q1.
>
> **Q3: Does the proposed geometric reconstruction module handle self-occlusion?**
>
> A3: Unfortunately, our reconstruction module cannot fully address self-occlusion.  Our method is based only on a 2D reconstruction, and it is difficult to reason out occlusions without multi-view or 3D information. However, we are able to outperform pseudo-labelling strategies in many challenging occlusion cases. One example is given in Supplementary Fig. A, where pseudo-labelling mixes up the left and right-ankles, while we can prevent such inversions since we explicitly link child to parent nodes in the reconstruction.
>
> Thanks for all these interesting and insightful questions. We have added clarifications from our answers to the revision.
>
> **Q4: Some details of comparison experiment maybe needed, i.e. whether the methods listed in table 1&2 use the same backbone network? is the computational cost and memory cost comparable, etc.**
>
> A4: Thank you for pointing this out. Yes, the methods listed in table 1&2 all use the same backbone network, i.e. ResNet101. The computation cost and memory cost during inference will be the same. We will clarify them in the revision.

---

> > ### Comment · Reviewer_RVKB · 2023-08-18
> >
> > Thanks the authors for the answers to my questions. I don't have any further questions.

---

### Official Review · Reviewer_kmmR · 2023-07-06

**Soundness:** 3 good
**Presentation:** 2 fair
**Contribution:** 3 good
**Rating:** 5
**Confidence:** 4

**Summary:**

This paper introduces a self-supervised synthetic-to-real domain adaptive pose estimation method. The approach utilizes a reconstruction-based strategy. It involves geometric transformations with an image generation network to reconstruct a driving image from a base image and uses pixel-wise reconstruction losses to ensure accurate keypoint localization. The method also incorporates an integrated pseudo-labelling technique that combines confidence masks from 2D heatmap and 3D predictions, resulting in more usable pseudo-labels. The performance of the approach is assessed using evaluation metrics including Percentage of Correct Keypoint (PCK) and end-point error (EPE). The study includes ablation studies as well as experiments in 2D hand/human keypoint detection and 3D hand pose. The experimental results demonstrate improvements over the selected state-of-the-art methods in terms of the main evaluation metrics.

**Strengths:**

- The paper ist well-structured.
- The incorporation of meaningful ablation studies strengthens the scientific validity.

**Weaknesses:**

- StyleNet was not adequately explained.

Table 1:
- It was not clarified that MCP, PIP, and DIP refer to the joints of the hand.
- Although PIP and DIP in the MVHand dataset for the proposed method do not perform better than UniFrame, Fin in the proposed method outperforms the Fin in UniFrame. The reason for this improvement is unclear.

Table 2:
-The discussion regarding the results on the 3DPW dataset in comparison with the Human3.6M dataset was not sufficient.

- It would be highly beneficial to provide more visualizations of the final results, at least in the supplementary material, particularly for the 3DPW dataset.

The explanation regarding the comparison of EPE in Table 5 (b) was insufficient.

Figure A in the supplementary:
For Human3.6M, the head in UniFrame was detected much more accurately than in the proposed method (ours), contrary to the expectation based on Table 2, where the proposed method was expected to have better head detection.

It would enhance clarity if the paper provided a brief explanation of the state-of-the-art methods, particularly UniFrame. This would allow for a better understanding of the unique contributions of the paper in comparison to its competitors.

**Questions:**

How many frames does the 3DPW dataset consist of?

**Limitations:**

The potential societal impact is not discussed.
It would be beneficial to include more visualized successful cases in the comparisons with state-of-the-art methods. For instance, in Figure A for Human3.6M, it is observed that the proposed method, contrary to the expectation based on Table 2, suffers from the same hip problem as other state-of-the-art methods.

---

> ### Author Rebuttal · Authors · 2023-08-08
>
> We greatly appreciate you highlighting the parts of our paper which could benefit from more explanation and discussion.  We will definitely include the suggested points in the revision and hope it will help you view our work more positively.
>
> **Q1: Explanation of StyleNet.**
>
> A1: Thanks for pointing out this oversight. For style transfer, we follow UniFrame [2] and use Aladin [1]. The style transfer network is an encoder-decoder that adjusts the first and second-order moments in the adaptive instance normalization layers to transfer the target style to a source image while maintaining source semantics. We will add details to Section 3.2.
>
> [1] Aladin: all layer adaptive instance normalization for fine-grained style similarity, ICCV’21
> [2] A Unified Framework for Domain Adaptive Pose Estimation, ECCV’22
>
> **Q2: MCP, PIP, and DIP are joints of the hand.**
>
> A2: Thanks for pointing out this oversight. MCP, PIP, DIP and Fin refer to the MetaCarpoPhalangeal joints (MCP), Proximal InterPhalangeal joints (PIP), Distal InterPhalangeal joints (DIP), and hand fingertips (Fin). We will clarify it in the final version.
>
> **Q3: Why proposed method outperforms UniFrame for fingertips but not PIP and DIP joints.**
>
> A3: Thanks for pointing out this curious phenomenon. Our approach relies on the reconstructed image to serve as a supervisory cue. The reconstruction is less sensitive to intermediate joints like DIP and PIP as these can often compensate for each other. The reconstruction is more sensitive to endpoint locations, i.e. the fingertips. If the predicted fingertip is located on the background, then upstream segments will feature background pixels in the reconstruction. If the predicted fingertip is located on the hand but is incorrectly placed, certain foreground pixels will be unaccounted for. Both cases will be penalized in the reconstruction loss.
>
> We will add this explanation to the revision.
>
> **Q4: More discussion on 3DPW results vs. Human3.6M. It would be highly beneficial to provide more visualizations of the final results, at least in the supplementary material, particularly for the 3DPW dataset.**
>
> A4: Thank you for pointing this out. We added visualizations from 3DPW to the rebuttal pdf and will include more samples in the revised supplementary. Since 3DPW is collected in the wild, it is more challenging and features more complexity in pose, appearance and background compared to Human3.6M. Our paper is the first to verify that adaptation is feasible on 3DPW (previous approaches only target Human3.6M) and highlights the strength of our approach.
>
> **Q5: The explanation regarding the comparison of EPE in Table 5 (b) was insufficient.**
>
> A5: Thanks for your suggestion. We will add more explanations in the revision.
>
> Table 5(b) shows that all methods can reduce the domain gap to some extent. SemiHand, using pseudo-labelling and consistency learning, has improvements of 8.58 mm and 1.46 mm on the H3D and MVHand, respectively. DualNet builds on the framework of Semihand and adds depth inputs for additional improvements of 2.11 mm and 3.3 mm respectively.
>
> Our work, using only RGB inputs, emphasizes the benefits of reconstruction as a supervisory signal. Even without consistency or depth inputs, we can outperform SemiHand and DualNet with improvements of 10.85 mm and 4.84mm. This result highlights that 3D pose estimation can benefit from 2D reconstruction. Results of Fig. 3(a) show qualitatively that reconstruction complements pseudo-labelling, as it refines end-point errors such as the fingertips.
>
> **Q6: Figure A in the supplementary: For Human3.6M, the head in UniFrame was detected much more accurately than in the proposed method (ours), contrary to the expectation based on Table 2, where the proposed method was expected to have better head detection.**
>
> A6: Thanks for pointing out this inconsistency. Compared with Uniframe, our performance on the head is indeed much more competitive (63.5 vs. 86.5). We presented more representative samples in the rebuttal pdf, and we will provide more visualizations in the Supplementary.
>
> **Q7: It would enhance clarity if the paper provided a brief explanation of the state-of-the-art methods, particularly UniFrame. This would allow for a better understanding of the unique contributions of the paper in comparison to its competitors.**
>
> A7: Thanks for your suggestion. The state-of-the-art methods, including UniFrame, CCSSL, RegDA, focus on the pose space to directly refine the predicted keypoints. Specifically, for UniFrame, it uses style transfer to augment source data and align the predictions from the student model to the pseudo-labels of the teacher model.
>
> In contrast, we focus on improving the keypoints indirectly through image reconstruction. Such an approach is less sensitive to annotation gaps and is especially effective for accurate endpoints.
>
> **Q8: How many frames does the 3DPW dataset consist of?**
>
> A8: 3DPW contains 23543 frames for training and 35515 frames for testing.
>
> **Q9: The potential societal impact is not discussed.**
>
> A9: We do not foresee direct negative societal impacts from our work, though accurate human/hand keypoint estimation may facilitate downstream tasks such as creating deep fakes.  We will urge readers to make ethical use of our work in the revision.
>
> **Q10: Include more visualized successful cases in the comparisons with state-of-the-art methods. For instance, in Figure A for Human3.6M, it is observed that the proposed method, contrary to the expectation based on Table 2, suffers from the same hip problem as other state-of-the-art methods.**
>
> A10: Thanks for your suggestion. We have added more successful cases in the rebuttal pdf and will include more in the revised supplementary material. The hip problem arises from differences in annotation convention so all methods (including ours) have lower accuracy on the hips. However, we outperform others by 15.4% on Human3.6M.

---

> > ### Author Response · Authors · 2023-08-20
> >
> > Thanks again for your valuable suggestions. We sincerely hope you've had a chance to review our rebuttal, as we believe our response adequately addresses your concerns. Since the discussion period is ending soon, please let us know if there are any questions or concerns. If we've covered your concerns, we would greatly appreciate your consideration in raising your score.

---

### Official Review · Reviewer_QXSS · 2023-07-22

**Soundness:** 3 good
**Presentation:** 3 good
**Contribution:** 3 good
**Rating:** 5
**Confidence:** 3

**Summary:**

This paper addresses the task of learning human body and hand pose estimation from labeled synthetic data and unlabeled data of the target domain. The main idea is to improve the key point estimation pre-trained on synthetic data by reconstructing a video frame using another frame with key point estimation, affine transformation, optical frame estimation, and image synthesis. This paper also proposes a novel pseudo-labeling method using 2D head-map-based method [16] and 3D pose correction [41] jointly. The experiments demonstrate that the proposed method outperforms existing syn-to-real methods on several hand and human body datasets, and the effectiveness of the proposed two components (reconstruction and pseudo-labeling).

**Strengths:**

- This paper proposes a method to exploit unlabeled data by image reconstruction with predicted keypoints. Since the most existing method in this task is to use pseudo-labeling, this method is in a new direction and the novelty would be high.
- A novel pseudo-labeling strategy using 2D confidence [16] and 3D confidence with pose correction [41]. Although the proposed method is rather simple, it outperforms the SOTA pseudo-labeling method [16].
- State-of-the-art accuracy on multiple benchmarks when the proposed reconstruction and pseudo-labeling are used together. It is also good that the two proposed methods (reconstruction and pseudo-labeling) are not only complementary, but rather reinforce each other. It has been shown that reconstruction improves the ratio of valid pseudo-labels.
- The proposed method significantly improves the accuracy of hard semantic categories where annotation gap exists that existing methods do not work very well.


**Weaknesses:**

- There are several reconstruction-based methods [38,11,29,17] as described, but no experimental comparison has been provided, and the proposed method may be inferior. Is there any reason why these methods cannot be compared with the proposed method?
- It is clearly demonstrated that the accuracy of the new technique is superior. However, the underlying mechanism seems less clear from the manuscript. In particular, why is the annotation gap between synthetic and real datasets overcome? What (and how) information or inductive bias is obtained in the reconstruction strategy?
- Unlike existing methods, the proposed method requires a video dataset of the target domain. (However, I think this is a minor issue because this kind of data is easy to obtain).


**Questions:**

- Please see also the weakness section.
- Where did the "12%" in the abstract/introduction come from? This number does not appear in the experiment section.
- Is invariant feature learning (p. 3.2) an original contribution? Or are there similar methods?
- The curve in Table 3 (b) looks inconsistent with the description in the main body. Maybe the legends are reversed?


**Limitations:**

- Please see the weakness section.

---

> ### Author Rebuttal · Authors · 2023-08-08
>
> **Q1: Experimental comparisons to reconstruction-based methods [38,11,29,17].**
>
> A1: Thanks for your suggestions. We added experiments on [29] and opted to exclude others due to the inapplicable rendering [38] or additional requirements like 2D labels [11, 17].
>
> Specifically, the rendering in [38] is designed for depth maps and is not applicable to our RGB task. Rendering RGB images is still challenging. For [11] and [17],  they both require additional information, such as unpaired or paired ground truth 2D, which gives them an unfair advantage. Instead, we only model the geometric transformations without other explicit supervision on the target domain.
>
> We compared against [29] and will add the result to the revision. We find ours significantly outperforms [29] because [29] relies on a pre-defined template (front pose) to predict pose transformation, leading to failure on behind poses, or folded body poses, as shown in the bottom half of rebuttal pdf.
>
>
> **Q2: Underlying mechanism of improved accuracy. Why is the annotation gap between synthetic and real datasets overcome? What (and how) information or inductive bias is obtained in the reconstruction strategy?**
>
> A2: One gap comes from different annotation conventions -- for example, the hip keypoints in SURREAL are closer to the pelvis than Human3.6M, even though the two datasets use the same skeleton topology (refer to Fig. B in Supplementary). Methods that use keypoints directly for supervision, e.g., pseudo-labels, will therefore be more sensitive to this difference.  In our case, our supervisory signal comes from both the reconstructed image and the pseudo-labelling. Specifically, the proposed reconstruction refines the keypoints according to the shared skeleton topology and tends to satisfy both datasets. Therefore, it’s less sensitive to annotation differences and will help improve the keypoint predictions like hips (refer to the results in Table 2).
>
> **Q3: Unlike existing methods, the proposed method requires a video dataset of the target domain. (However, I think this is a minor issue because this kind of data is easy to obtain).**
>
> A3: In principle, our method requires only two images with the same target instance (human or hand).  However, diverse backgrounds and very different poses may present additional challenges so we advocate the use of video.
>
> **Q4: Where did the "12%" in the abstract/introduction come from? This number does not appear in the experiment section.**
>
> A4: Thanks for pointing out this oversight. We tabulate 12% by averaging the most significant improvements (fingertips on hands, shoulders on humans):
>
> * 8.5% (73.9 vs. 68.1) on the RHD->H3D
> * 5.9% (61.0 vs. 57.6) on the RHD->MVHand
> * 15.0% (89.8 vs. 78.1) on the SURREAL->Human3.6M
> * 18.7% (75.7 vs. 63.8) on the SURREAL->3DPW.
>
> We will add this tabulation to Section 4.4.
>
> **Q5: Is invariant feature learning (p. 3.2) an original contribution? Or are there similar methods?**
>
> A5: No, it is not an original contribution and we do not claim it as such. Invariant feature learning is standard practice for domain adaptation of classification [1], semantic segmentation [2]  and depth estimation [3]. Current works [4,5,6] in domain adaptation of pose estimation only utilize style transfer as data augmentations, we are the first to apply it to perform invariant feature learning.
>
> [1] Feature Stylization and Domain-aware Contrastive Learning for Domain Generalization, ACM MM’21
>
> [2] Learning Texture Invariant Representation for Domain Adaptation of Semantic Segmentation, CVPR’20
>
> [3] Real-Time Monocular Depth Estimation using Synthetic Data with Domain Adaptation via Image Style Transfer, CVPR’18
>
> [4] Enhancing Human Pose Estimation in Ancient Vase Paintings via Perceptually-grounded Style Transfer Learning, ACM Journal on Computing and Cultural Heritage’22
>
> [5] Sim2Real Instance-Level Style Transfer for 6D Pose Estimation, IROS’22
>
> [6] A Unified Framework for Domain Adaptive Pose Estimation, ECCV’22
>
> **Q6: The curve in Table 3 (b) looks inconsistent with the description in the main body. Maybe the legends are reversed?**
>
> A6: Thank you for pointing this out. Yes, they are reversed. We will correct them in the revision.

---

> > ### Comment · Reviewer_QXSS · 2023-08-15
> > **Post-rebuttal discussion**
> >
> > Dear Authors,
> >
> > Thank you for providing the author feedback. All my concerns have been well addressed. I think the justification about the annotation gap is reasonable (although the method still partially relies on keypoints), and the discussion about the comparison with [38,11,29,17] (explanation or additional experiment) is also convincing.
> >
> > Thank you for the thorough explanation. I have no more questions.

---

### Author Rebuttal · Authors · 2023-08-08

Dear reviewers and ACs,

Thank you for your careful and thoughtful reviews. R1 and R4 commented that the proposed geometric reconstruction for domain adaptive pose estimation is new and novel. All reviewers appreciated the empirical verifications to validate the effectiveness of the proposed methods.

---

We highlight the following to help understand our paper:

- As recognized by R1, our key novelty is the incorporation of image reconstruction for domain adaptive pose estimation -- existing works rely only on pseudo-labeling.

- The reconstruction complements pseudo-labelling by (1) emphasizing accurate endpoint placements (see R2 Q3 for details) (2) being less sensitive to different annotation conventions between source and target domains, e.g. shoulders for human (see R1 Q2 for details).

---

Since the concerns voiced by the reviewers are mainly non-overlapping, we offer detailed responses individually to each review. Also, we attach a pdf that contains the visualizations for the reviewers to refer to.

---

### Decision · Program_Chairs · 2023-09-21

**Decision:**

Accept (poster)

**Comment:**

The paper seek to address the challenge of sim-to-real transfer in human body pose estimation. The initial reviews were mixed. The reviewers raised concerns on the clarity of technical details, as well as the positioning w.r.t. to prior methods. The authors provided a detailed response. The reviewers were satisfied with the response, resulting in overall positive final ratings.

After going through the paper, the review, and the response, the AC values the key idea of the paper (considering geometric reconstruction for domain adaptation in pose estimation), and recommends the acceptance of the paper. The AC also recognizes that the technical components in the current manuscript were not particularly well described, and thus urges the authors to further improve the clarity of the presentation.